# The Comparison of InSb-Based Thin Films and Graphene on SiC for Magnetic Diagnostics under Extreme Conditions

**DOI:** 10.3390/s22145258

**Published:** 2022-07-14

**Authors:** Semir El-Ahmar, Marta Przychodnia, Jakub Jankowski, Rafał Prokopowicz, Maciej Ziemba, Maciej J. Szary, Wiktoria Reddig, Jakub Jagiełło, Artur Dobrowolski, Tymoteusz Ciuk

**Affiliations:** 1Institute of Physics, Poznan University of Technology, Piotrowo 3, 61-138 Poznan, Poland; marta.przychodnia@put.poznan.pl (M.P.); jankowski.kuba@gmail.com (J.J.); maciej.szary@put.poznan.pl (M.J.S.); wiktoria.reddig@student.put.poznan.pl (W.R.); 2National Centre for Nuclear Research, Andrzeja Soltana 7, 05-400 Otwock, Poland; rafal.prokopowicz@ncbj.gov.pl (R.P.); maciej.ziemba@ncbj.gov.pl (M.Z.); 3Łukasiewicz Research Network—Institute of Microelectronics and Photonics, Aleja Lotnikow 32/46, 02-668 Warsaw, Poland; jakub.jagiello@imif.lukasiewicz.gov.pl (J.J.); artur.dobrowolski@imif.lukasiewicz.gov.pl (A.D.); tymoteusz.ciuk@imif.lukasiewicz.gov.pl (T.C.)

**Keywords:** Hall effect sensor, InSb, graphene, magnetic diagnostics, neutron irradiation, radiation-resistant materials, self-healing effects

## Abstract

The ability to precisely measure magnetic fields under extreme operating conditions is becoming increasingly important as a result of the advent of modern diagnostics for future magnetic-confinement fusion devices. These conditions are recognized as strong neutron radiation and high temperatures (up to 350 °C). We report on the first experimental comparison of the impact of neutron radiation on graphene and indium antimonide thin films. For this purpose, a 2D-material-based structure was fabricated in the form of hydrogen-intercalated quasi-free-standing graphene on semi-insulating high-purity on-axis 4H-SiC(0001), passivated with an Al_2_O_3_ layer. InSb-based thin films, donor doped to varying degrees, were deposited on a monocrystalline gallium arsenide or a polycrystalline ceramic substrate. The thin films were covered with a SiO_2_ insulating layer. All samples were exposed to a fast-neutron fluence of ≈$7\times 10^{17}$ cm^−2^. The results have shown that the graphene sheet is only moderately affected by neutron radiation compared to the InSb-based structures. The low structural damage allowed the graphene/SiC system to retain its electrical properties and excellent sensitivity to magnetic fields. However, InSb-based structures proved to have significantly more post-irradiation self-healing capabilities when subject to proper temperature treatment. This property has been tested depending on the doping level and type of the substrate.

## 1. Introduction

Magnetic field diagnostics have always been discussed in the context of cross-cutting technologies that enable the use of specific devices and components in space and on Earth. However, in recent years, the problem has received increased interest, following the rise of high-temperature electronics [1,2,3,4] and electronics for extreme environments [5]. Applications for extreme environments include the defense, oil, gas, automotive, aerospace, and geothermal industries. This dictates the need for continuous development of new types of materials or the exploration of the properties of the existing materials, in directions that have not been verified so far. Extreme-environment electronics have been strongly promoted, especially in the context of the energy industry and towards the use of controlled nuclear fusion for global energy supplies.

The ongoing efforts demand a complex sensor infrastructure necessary for monitoring thermonuclear processes [6,7]. One of the crucial diagnostic elements is magnetic-field sensors, which are required for efficient and sustainable confinement of plasma. This necessitates the sensors to operate effectively for extended periods under high temperatures and highly destructive neutron radiation (NR). However, this has proven an extremely demanding task. Hence, research into optimal sensing platforms continues.

The planned fusion installations are known to differ. Hence, the exact conditions the sensors will have to operate under will vary. Most of the investigated solutions employed Hall-effect-based sensors. However, maintaining a good sensing performance with sufficient stability in such difficult conditions has remained a challenge for over a decade. Current research on materials capable of stably detecting magnetic fields under extreme conditions in tokamaks focuses mainly on thin films made of chromium [5], bismuth [8,9,10], gold, and antimony [11,12], but III-V compound semiconductors have also been investigated [13,14,15]. Furthermore, recent reports have also considered the viability of two-dimensional (2D) materials in the form of quasi-free-standing (QFS) epitaxial Chemical Vapor Deposition (CVD) graphene on semi-insulating silicon carbide (GR/SiC) [16] or CVD graphene transferred onto a sapphire substrate [17].

This investigation aims to compare the retention of sensing characteristics of indium antimonide (InSb) and QFS GR/SiC under neutron irradiation. InSb thin films can be used to manufacture signal-type Hall effect sensors, as well as very precise measurement-type Hall effect sensors [18]. Their capability to work under extreme conditions has been demonstrated for an extremely wide temperature range (from −270 °C to 300 °C) [19] and NR [14]. In addition, the technology of InSb-based structures has been widely developed. The usefulness of the GR/SiC platform in magnetic field detection has been verified, and its potential for stable work at temperatures as high as 500 °C has also been proven [4,20]. Furthermore, its resistance to NR was only recently indicated [16].

In this paper, we present an experimental comparison of the effect of NR on a 2D material in the form of QFS GR/SiC and on III-V compound semiconductor thin films in the form of InSb-based systems. We report on the resistance of the structures to NR, the effects of NR on their electrical parameters, the mechanisms by which the electrical characteristics are affected, and the possible self-healing mechanisms.

## 2. Materials and Methods

### 2.1. Samples Preparation

In total, ten samples of various types of Hall effect structures were prepared for the experiment. All of them could be grouped into four categories. Figure 1 illustrates an optical image of the individual representatives of each category.

The first category groups InSb-based thin films on a polycrystalline substrate (Sitall) (Figure 1a). The second one represents thin films based on InSb epitaxially grown on semi-insulating gallium arsenide (i-GaAs) (Figure 1b). The third group is related to InSb-based high-temperature Hall-effect sensor structures (HTHS) (Figure 1c). The fourth one is represented by a single graphene-based HTHS (Figure 1d).

All InSb-based structures were fabricated using the flash evaporation technique [21]. Five of the InSb thin films were evaporated onto an insulating Sitall substrate, where one of the thin films was left undoped and the other four were doped with Te or Se to the level of the electron density *n* between $3.0\times 10^{17}\;{\mathrm{cm}}^{-3}$ and $1.1\times 10^{18}\;{\mathrm{cm}}^{-3}$ (1.7–3 μm thick). Four of the InSb thin films were epitaxially grown on an i-GaAs substrate. Of these, one sample was undoped and three were doped with Sn to the level of *n* between $1.6\times 10^{18}\;{\mathrm{cm}}^{-3}$ and $2.9\times 10^{18}\;{\mathrm{cm}}^{-3}$ (1 μm and 4.5 μm thick). The InSb-based HTHS was fabricated as described in Refs. [19,22]. The InSb active layer was shaped in the form of a cross to precisely measure the Hall effect. The structures were equipped with metallic Cr/Au electrodes (100 nm/300 nm) and coated with a 100 nm insulating SiO${}_{2}$ protective layer. Detailed information on the fabrication processes is included in Refs. [23,24,25].

QFS graphene [26,27,28,29] necessary for the graphene-based HTHS was grown on semi-insulating high-purity on-axis 4H-SiC(0001) (Cree Inc.) in a hot-wall Aixtron VP508 reactor using the epitaxial Chemical Vapor Deposition method [30], with thermally decomposed propane as a carbon source and in situ hydrogen atom intercalation [31]. The substrate was processed into a 1.6 mm × 1.6 mm Hall effect structure featuring a cross-shaped [32] 100 μm × 300 μm graphene mesa and four Ti/Au (10 nm / 60 nm) ohmic contacts, all passivated with a 100 nm-thick aluminum oxide (Al_2_O_3_) layer synthesized from trimethylaluminum (TMA) and deionized water at 670 K in a Picosun R200 Advanced Atomic Layer Deposition (ALD) reactor [33,34]. Detailed information on the fabrication processes is included in Refs. [16,20].

### 2.2. Fast Neutron Irradiation

The NR process was carried out at the National Centre for Nuclear Research, Poland, in the MARIA research nuclear reactor [35]. A fast neutron irradiation facility was used for this purpose. The facility is located at the outskirts of the MARIA reactor core. Therefore, the radiation conditions are in the facility not as severe as in the core—fast neutron flux and nuclear heating are more than one order of magnitude lower, but thermal (slow) neutron flux is more than three orders of magnitude lower than in the central part of the reactor core. Thus, the facility is perfect for testing and qualifying components and systems intended for use in fusion devices. In Figure 2, the energy spectrum in the facility (blue) is compared with a representative example of the spectrum in the central, usable part of the core (red).

The fabricated samples were exposed to high-energy neutron radiation with a fluence of ≈$6.6\times 10^{17}{\mathrm{cm}}^{-2}\pm 5\%$. The samples were irradiated in three independent processes while maintaining similar process parameters (the difference is included within the error of the applied NR fluence). Each of the processes followed an identical procedure. Each batch of structures was secured and immobilized in a small quartz container using quartz wool. Then, it was tightly closed and filled with helium in a larger metal container. Finally, the metal container was placed in the appropriate position in a collective container. The position of the samples inside the collective container was fixed in such a way as to obtain the maximum vertical distribution of the fast neutron flux density during the irradiation. The collective container was placed in one of the channels of fast neutron irradiation facility in the MARIA reactor. Detailed information on the irradiation process can be found in Ref. [16].

### 2.3. Determination of the Electrical Parameters

The estimation of the Hall-effect-derived electrical parameters was performed based on the *van der Pauw* model [36]. The measurements of the structures presented in Figure 1 were carried out according to the following procedure. InSb-based structures in the six-terminal version (sample no. *I1-I8*, see Figure 1a,b), were designed for measurements using terminals *1* and *3* for the injection of the driving current *I*. Terminals *4* and *6* were used to read the voltage drop $U_{R}$, from which the sample resistance $R=(U_{R}/I)$ was determined. The Hall voltage $U_{H}$ was measured between terminals *2* and *5*. Measurements of $U_{R}$ and $U_{H}$ were made by connecting wires from electrodes *1-6* to a suitable measurement handle.

InSb-based HTHS (sample no. *I9H*, see Figure 1c) is a Maltese-cross-shaped structure, and the graphene-based HTHS (sample no. *G1H*, see Figure 1d) is an isosceles cross. Both structures were equipped with four symmetrically located electrical terminals. Connecting the driving current *I* and measuring $U_{H}$ and $U_{R}$ can take place on any opposite pair of terminals. The resistance *R* was measured between the two opposing electrodes and $U_{H}$ between the other two opposite electrodes.

From these measurements, the carrier density *n* of InSb films (or sheet carrier density $n_{s}$ for the 2D active layer) was determined using the following formulas:(1)$$n=\frac{IB}{qdU_{H}},$$
(2)$$n_{s}=\frac{BI}{qU_{H}},$$
where *B* is the value of the applied magnetic induction, *q* is the unit charge, and *d* is the film thickness. The charge carrier mobility μ was determined using:(3)$$\mu =\frac{U_{H}}{IBR}\frac{l}{w},$$
where l/w is the active layer length-to-width ratio. For the purposes of the calculations, the ratio l/w=4 was established for *I1–I8* samples, l/w=2.5 for the *I9H* sample, and l/w=3 for the *G1H* sample.

The data on μ and *n* (or $n_{s}$, in case of 2D layer) of the fabricated structures are collected in Table 1. Table 1 is broken down into four sections: after the thermal stabilization (*A*), after the irradiation (*B0*), and after the post-NR temperature treatment up to $300{\;}^{{}^{\circ}}$C (*B1*) and $350{\;}^{{}^{\circ}}$C (*B2*). The InSb-based samples were numbered *I1* to *I9H*, arranged by an increasing initial *n*. The single graphene-based sample is denoted *G1H*. Table 1 also contains information on the dopant type, substrate type, and the thickness of the InSb thin films.

### 2.4. Temperature Treatment

After the NR, the samples were subjected to a temperature treatment (up to $350{\;}^{{}^{\circ}}$C). In general, the temperature treatment strategy was as follows. It was carried out in cycles, with the maximum temperature of each cycle $50{\;}^{{}^{\circ}}$C higher than the previous one. Each temperature cycle was combined with annealing. The annealing time during each cycle did not exceed 1 h, and the Hall effect measurements were made during the annealing every 5 to 10 min. The exact annealing time (individual for each sample) was dependent on changes in the electrical parameters of a particular sample. When changes in the electrical parameters were negligibly small during annealing (or significant but stable during some part of the annealing process), the sample was cooled down to room temperature (RT), and another cycle was carried out. An example of the complete temperature treatment process of the *I6* structure is shown in Figure 3.

According to the above-mentioned procedure, the *I6* was subjected to four successive temperature cycles as part of the temperature treatment. The measurement of the electrical parameters during each cycle is shown in Figure 3. The summary of the final parameters after each successive cycle of the temperature treatment procedure of the *I6* sample is presented in Figure 4a. For comparison, Figure 4 shows a similar summary for the *I7* sample (monocrystalline). Due to a large amount of data and the fact that most of the changes in the electrical parameters of the samples took place during (or after) the annealing at $300{\;}^{{}^{\circ}}$C, Table 1 contains only the results of the Hall effect measurements after the annealing at $300{\;}^{{}^{\circ}}$C and $350{\;}^{{}^{\circ}}$C.

## 3. Results and Discussion

### 3.1. Type of Damage Induced by Fast Neutrons in the Investigated Materials

Unlike thermal neutrons, which are responsible for transmutations, fast neutrons mainly induce the formation of various kinds of crystal lattice defects of InSb. The observed changes occur mainly through the trapping of charge carriers in these defects [37]. Depending on the energy of the fast neutron, in a single collision act between the fast neutron and the crystal atom, $10^{3}$–$10^{6}$ lattice atoms are heated to very high temperatures ($10^{4}$ K) and rapidly quenched (10–11 s). Thus, defects in the area are frozen [38]. Many types of lattice disturbances caused by fast neutron bombardment are postulated, e.g., thermal spikes, displacement spikes, plasticity spikes, replacement collisions, and focusing collisions, and transmutation processes where silver, tin, tellurium, and cadmium are created [39]. Disordered regions may have quite a different lattice structure and electrical parameters compared to non-disturbed regions. It is estimated that if these areas are approximately spherical, the radius will be (150–$200\;\overset{˚}{A}$). Such regions may also have different types of conductivity, which generally create of two types of defects: acceptor and donor type [40]. When the energy of the hitting atom is sufficiently low, the distance between the next collisions, where large energy is lost, approaches interatomic distances. According to various models, the damaged area consists of a region that contains a very high concentration of vacancies, surrounded by a region with a high concentration of interstitial atoms [37].

The modification of the Al_2_O_3_/GR/SiC system should be considered in a slightly different way. Here, the active layer is a 2D carbon structure formed on a SiC substrate, the (0001) surface of which is saturated with hydrogen atoms. Contrary to the thin-film structures, in the case of GR/SiC, the damage can be located outside of the active layer. As neutrons interact only with the atomic nuclei, if the interaction is a neutron scattering on the nucleus, in extreme situations it may result in knocking the entire atom out of its nodal position. If a neutron enters the nucleus, it will promote some specific reaction that will change the nucleus into another isotope of the given element, or another element, while emitting electromagnetic or corpuscular radiation. This usually has consequences for the entire crystal lattice of a given system.

Defects in QFS graphene that could affect the *sp^2^* hybridization of the C-C bonding are nuclear transmutation, fast neutron scattering, interactions with the substrate, and interdiffusion. Among the chemical elements used in the GR/SiC system, carbon has the highest cross-section for the scattering of neutrons, but only for thermalized neutrons, i.e., those with low energy. Carbon has a fast neutron cross-section of $2.3\times 10^{-24}\;{\mathrm{cm}}^{2}$. Thus, the scattering of fast neutrons is significantly more likely to directly affect QFS graphene. Interactions with the substrate facilitated by exposed patches of SiC(0001) can affect the structure of QFS graphene. This effect depends on the level of intercalation and the distribution of hydrogen atoms.

### 3.2. The Effect of NR on the Electrical Parameters of InSb and GR/SiC

Figure 5 shows the carrier density ($n/n_{s}$) and mobility (μ) of InSb and GR/SiC before irradiation (black), after irradiation (red), and after the complete temperature treatment (gray). The left-hand side of Figure 5a shows carrier-density values of InSb-based structures (*I1*–*I9H*) in an ascending order of the initial *n*. An inset shows the values in finer detail for samples *I1*–*I5*. The right-hand side of Figure 5a presents the sheet carrier density $n_{s}$ of the graphene sample (*G1H*).

Figure 5b shows the values of carrier mobility. This figure uses the same composition as Figure 5a. The comparison between InSb and GR/SiC is intended to illustrate the relative degree of change in both materials. The corresponding parameters are given per cm^3^ and cm^2^, respectively. Thus, the values cannot be directly compared.

Figure 5b proves that neutron radiation had a notable effect on the values of μ. On average, μ has decreased by 83 % (*I1*–*I9H*). However, no significant differences were found between the samples. The average decrease in mobility for InSb on monocrystalline substrates was 80 %, and for the polycrystalline ones, it was 86%. In contrast, Figure 5a shows that neutron radiation can have a varied impact on the carrier density in InSb. In some cases, the change is relatively small (e.g., *I5*), while for other samples, the effect is significant (e.g., *I9H*).

The character of the changes correlates with the initial value of *n*. If *n* is initially low ($n<3.7\times 10^{17}\;{\mathrm{cm}}^{-3}$), the irradiation tends to increase the concentration (see the values for samples *I1*–*I4*). However, if the initial *n* is high ($n⩾3.7\times 10^{17}\;{\mathrm{cm}}^{-3}$), the effect reverses (samples *I5*–*I9*). Similar trends were reported by *Cleland* [38], where under fast neutron radiation, p-type InSb changed its conductivity to n-type, and for n-type samples, the concentration decreased.

The results suggest a phenomenon comprising two competing processes in which irradiation can produce both donor and acceptor defects. Furthermore, the ratio between both appears to depend on the concentration of atomic impurities introduced to the InSb crystal lattice during the evaporation process, as it determines the value of *n*. The dopants should have a negligible impact on transmutations in the crystal, as the concentration values are low. Thus, this cannot be the mechanism behind the variation in post-irradiation carrier density. Instead, if the doping level is low, the effect is likely dominated by lattice distortion, which produces higher-energy configurations that tend to facilitate fewer bonds by some of the affected atoms. Unsaturated electron pairs could introduce new donor levels, thus increasing the initial electron concentration. On the other hand, if the doping level is high, the effect should also occur. However, it must compete with another process that affects donor states introduced by the dopants. In this case, the structural damage sustained by the samples likely causes the impurities to occupy inactive places, creating electron traps. This reduces the initial contribution of dopants to the charge concentration. Thus, the effect is minor in samples with low impurities.

Interestingly, the unique characteristics of the competing processes should allow one to match the doping level and the irradiation dose to compensate for the change in *n*. This concept is best illustrated by samples *I4* and *I5*, where the relative changes in *n* are the smallest, and the trend reverses (see Figure 5a). *I4* has an initial *n* of 3.0 × 10^17^ cm^–3^, and after irradiation, the concentration increases to 4.7 × 10^17^ cm^–3^, which represents a relative change of 57%. In the case of *I5*, the initial *n* is 3.7 × 10^17^ cm^–3^, and after the irradiation, it decreases to 2.9 × 10^17^ cm^–3^, thus resulting in a −22 % relative change. Consequently, we can assume that for the initial value of *n* between 3.0 and 3.7 × 10^17^ cm^–3^, there is an equilibrium in the production of donor- and acceptor-type defects, which should pin the value of *n* for the irradiation dose we have used. Therefore, an initial carrier concentration of ≈3.5× 10^17^ cm^–3^ could prove the best for Hall sensors based on InSb thin films when designed to operate under a fast neutron radiation fluence of up to ≈$7\times 10^{17}$ cm^–2^.

In contrast, the graphene sample exhibits a relatively low degradation of the electrical parameters compared to the majority of InSb structures. *G1H* has an initial μ and $n_{s}$ of ≈900 cm^2^/(Vs) and 1.7 × 10^13^ cm^–2^, respectively. After the irradiation process, these values decrease to ≈500 cm^2^/(Vs) and 1.0 × 10^13^ cm^–2^, respectively, representing 43 and 39% change. This makes the relative reduction in mobility approximately half the average decrease for InSb (83 %). Regarding the carriers concentration, only sample *I5* is less affected than *G1H* (22 compared to 39%). For the remaining samples, the value of *n* is more affected, with its extent varying significantly.

As such, the change in the parameters of any of the InSb samples is not consistent with that of GR/SiC, suggesting that the materials are affected by fast neutrons differently. However, in this case, the disparities will not be a simple product of different cross-sections for fast neutrons of the elements comprising both materials. InSb samples are thin—but three-dimensional (3D)—strongly homogeneous structures, whereas GR/SiC is a layered heterostructure, in which the sensing layer is a 2D sheet. Consequently, the sustained damage will accumulate differently.

The atomic structure of InSb consists of two sublattices with four atoms (two In and two Sb) per 273 Å^3^ unit cell, which gives 2.9 × 10^22^ atoms/cm^3^. Each atom has a collision probability determined by its cross-section for neutrons and the fluence. Here, it is 3.2 × 10^–6^ for In and 3.5 × 10^–6^ for Sb. In combination, the values can be used to determine the density of primary knock-on atoms (PKAs), which, given the fluence, is 9.8 × 10^16^ atoms/cm^3^. This represents the direct damage to InSb. However, it is not the entire damage sustained by the structure, as PKAs have the kinetic energy to displace other atoms, ultimately resulting in a displacement cascade. For 1–2 MeV neutrons, such cascades in InSb displace ≈90 atoms on average, as reported by Guenzer and Manning [41]. Therefore, in total, 0.3‰ of the atoms in the InSb structure have been displaced after the irradiation.

For GR/SiC, the effects of NR on Al_2_O_3_ and SiC have a limited impact on the sensing performance of the structure [16]. Rather, it was reported that the main effects influencing detection occur in the close vicinity of the graphene sheet. Hence, the discussion of the direct effect of neutrons can be limited to graphene. There are two carbon atoms per 5.24 Å^2^ surface unit cell of graphene. This results in 3.8 × 10^15^ atoms/cm^2^. Given the fluence and cross-section for fast neutrons of carbon, the collision probability is $1.5\times 10^{-6}$, resulting in a PKA density of $5.8\times 10^{9}$ atoms/cm^2^. However, PKAs of graphene are unlikely to initiate cascades in the sheet due to its 2D nature. It was reported that the primary damage to the graphene sheet in GR/SiC is 15 %, with the remaining 85 % being a product of secondary effects, such as interdiffusion and loss of hydrogen intercalation [16]. Consequently, after the irradiation of *G1H*, 0.0086‰ of atoms in graphene should have been affected. The higher the number of displaced atoms, the more likely it is that the electronic parameters of the material will be affected. Hence, this is likely the reason for the differences between InSb and GR/SiC.

### 3.3. Temperature-Induced Self-Healing of the Irradiated Systems

After the irradiation, a series of annealings was conducted according to the temperature treatment procedure presented in Section 2.4. Figure 6 shows the changes in the initial level of *n* and μ after NR and after the last two temperature cycles up to 300 °C and 350 °C.

For the InSb-based samples, the initial increase in this parameter is observed with the increase in the initial *n*. This increase applies to samples with $n<1.0\times 10^{17}\;{\mathrm{cm}}^{-3}$ (*I1*–*I3*). For samples with an initial $n>1.0\times 10^{17}\;{\mathrm{cm}}^{-3}$ (*I4*–*I9H*), a decrease in the *n* is observed after the full temperature treatment process in relation to the initial *n*. These changes are best shown in Figure 6a (the height of the gray bars relative to the 100% level). The best results of *n* parameter recovery were obtained for the *I4* sample, 73% recovery of initial *n* parameter value, after applying annealing cycles of up to 350 °C. This means that the optimal range of initial *n* value, estimated on the basis of pure radiation susceptibility analysis (see Section 3.2), also seems optimal from the point of view of susceptibility to post-radiation thermal recovery of the structure.

High temperatures tend to decrease a substantial number of post-irradiation defects in thin films. For all the considered InSb-based structures, an increase in μ can be observed, which accompanies the process of temperature treatment. This trend persists up to temperatures of about 300–350 °C. By analyzing Figure 6b, it can be concluded that for different InSb structures, different temperature limits of annealing cycles can be determined, for which the post-irradiation increase in μ is at its maximum. For the highest recovery of the μ, the optimal post-NR annealing temperature for samples *I1*, *I7*, *I8*, and *I9H* was 350 °C. In the case of samples *I2*, *I5*, *I6*, the optimal annealing temperature was 300 °C. It follows that various parameters of optimal annealing are closely related to the type of substrate used in the specific systems. The annealing processes of two representative samples are summarized in Figure 4, where the electrical parameters (determined based on measurements made at RT after the successive thermal cycles) are shown. Here, sample *I6* represents the thin-film polycrystalline samples, while sample *I7* represents the monocrystalline thin films.

During the temperature cycle, some defects are removed faster than others are created. At a specific temperature, there is an equilibrium of both processes, and then the situation is reversed. For example, the *I6* sample during the temperature cycle to 350 °C exhibits characteristics for heating and cooling (Figure 3). This means that these complex defects are permanent and cannot be overcome at this temperature. The course of the temperature dependence of μ in the case of polycrystalline thin films can be explained by grain boundary barriers, dislocations, sizes of grains, and evaporation conditions.

Analyzing the data from Figure 5 and Figure 6 for the *G1H* sample, one can notice a low impact of the temperature treatment on the electrical parameters of the sample relative to the InSb-based structures. The changes in $n_{s}$ and μ for the 2D material were only 9% and 3%, respectively, with respect to the value after irradiation. This gives total values of 70% and 60% residue of these parameters, respectively, from the initial values before irradiation. It was predicted that, for the irradiated QFS H-intercalated Al_2_O_3_/GR/SiC structure, temperatures above 200 ° C facilitate the diffusion of hydrogen atoms from areas with higher coverage to areas with less coverage. This effect can reduce the surface area, where intercalation is too low to support graphene separation [16]. Consequently, the electrical response of the system observed in high-temperature Hall-effect measurements is caused by the scattered hydrogen atoms, which limits the local interactions of graphene with the SiC substrate.

The temperature during irradiation can be roughly estimated at 200 °C or a little lower. Such conclusions can be drawn based on the behavior of μ during the temperature cycles. For most of the samples, changes in this parameter are observed during the cycle up to 200 °C. However, these changes are much less significant than changes resulting from annealing in the subsequent cycles. Therefore, it can be concluded that the slow changes in μ below 200 °C may result from the fact that shallower defects had already been removed during the irradiation process as a result of the high temperature of thin films (see Figure 4). This estimate is consistent with the rough temperature reading inside the container in which the samples were placed during the irradiation process in the MARIA reactor (130 ± 20 °C); however, the temperature of the samples could be somewhat different from the temperature inside the container.

### 3.4. Discussion of Different Sensor Platforms

Material research on the possible sensor platforms capable of operating under the conditions specific to fusion reactors has been carried out for more than two decades. Various materials have been and are currently considered, ranging from classic semiconductors, through semi-metal layers, to two-dimensional carbon structures. It is often difficult to compare the most promising results obtained by different research groups to highlight the advantages and limitations of a given system. Here, however, we would like to compare our research with others of similar nature, all within reasonable limits.

The basic requirement for the material candidates to detect magnetic fields under the conditions of future fusion reactors is their thermal stability. This parameter should be correlated with the possibly high value of the Hall coefficient (with minimum temperature dependence) within a possibly wide temperature range. The latest research in the field concerns materials that show the above characteristics in an operating temperature range exceeding 500 °C [5,42].

Figure 7 provides information on the HT properties of the most promising materials that have been tested by various research groups in the past decade, with the intent of use in fusion reactors. The data on the considered sensor platforms are compared to other systems based on the collective data contained in Refs. [5,42]. The current sensitivity is defined as $S_{I}=\left(\partial U_{H}/\partial B\right)/I$.

By analyzing Figure 7, it can be clearly stated that epitaxial graphene on 4H-SiC(0001) has signal stability in the full temperature range and the highest signal value. This system has a temperature stability coefficient $S_{I}$ of 0.02%/°C between room temperature and 300 °C, and 0.06%/°C between 300 °C and 500 °C [20], which defines it as highly stable for wide-temperature-range applications. The magnitude of $S_{I}$ is 3–5 orders of magnitude greater compared with metallic materials, and an order of magnitude greater in relation to InSb structures, which are widely considered an excellent material for the construction of magnetic field sensors for extreme temperatures [19]. Figure 7 indicates that the GR/SiC sensor platform is the most optimal for applications up to 500 °C. The only metallic materials capable of working in such a wide temperature range are thin films based on antimony, chromium, and molybdenum. Among them, the highest signal values are shown by antimony, but at the same time, their temperature dependence is high. Chromium behaves much better, showing higher temperature stability with signals of an order of magnitude higher than those in the case of Mo. However, compared to GR/SiC, both the thermal stability and the signal strength remain much lower. The Sb, Cr, and Mo layers were tested in the temperature range reaching 550 °C, and GR/SiC 500 °C. This upper range of the GR/SiC platform temperature tests is due to the limitations of the Hall-effect measurement system. It is expected that the actual operating temperature limit of the system is higher.

The comparison of the most promising systems in terms of their radiation resistance is not as obvious. This is due to the significantly different conditions of the experiments using various nuclear reactors. Table 2 summarizes the results of experiments from the last two decades, in which various types of materials were exposed to NR to assess the influence on their electrical properties. The results for individual materials were differentiated according to the type of nuclear reactor used in the experiment and the neutron radiation fluence. The table includes the change in sensitivity observed after irradiation ($S_{I}$). Furthermore, Table 2 also contains information on the optimal value of the carrier density of a given system (if determined), which guarantees the lowest susceptibility of the signal to changes caused by the radiation, as well as temperature during irradiation, $T_{irr}$.

Based on the literature data, thanks to which Table 2 was created, and the experimental results presented in Section 3.2, it is possible to make a preliminary comparison of classical thin film and 2D epitaxial graphene in terms of their susceptibility to neutron radiation. However, it should be noted that over the past decade, almost all publications have focused on thin-film materials. Currently, there are only few scientific reports dealing with experimental results related to the influence of neutron radiation on graphene [16,17,50]. Therefore, such a comparison at the moment is quite limited and may only indicate certain trends.

The research on thin-film InSb-based structures, with fast neutron fluence in the range of (0.1–3.1) $\times 10^{16}$, carried out in the IBR-2 reactor spectrum showed the significant stability of this material. The stability of the signal was demonstrated at a level that did not exceed <1%. In this range of the radiation dose, the optimal input carrier density was estimated in the range of (0.1–2.5) $\times 10^{18}$ cm${}^{-3}$. Relatively low neutron fluxes resulted in the temperature generated during the irradiation process below 70 °C. Because of the low temperature of the process, even weakly donor-doped InSb layers could have low coefficients of thermal stability.

The closest experiment to that presented in this article was carried out using the WWR-2 reactor (fluence $7.4\times 10^{17}$, the temperature during the irradiation of 110 °C). The conclusions presented in [48] are consistent with ours in terms of both the magnitude of the signal drop after irradiation and the estimated optimal input carrier concentration value.

In the LVR-15 reactor, extensive studies of InSb structures were also carried out within the scope of higher doses of radiation (0.1–1.3) $\times 10^{18}$ cm ${}^{-2}$. There, the optimal input carrier densities that guarantee the highest radiation resistance were not considered. The temperatures during the irradiation did not exceed 100 °C, while the sensitivity changes reached 30%. Of the remaining materials in Figure 7 only Bi, Au, and GR/Al${}_{2}$O${}_{3}$ can reasonably be compared with the results of this study. In other cases, there are no clear data to enable the comparison. The neutron irradiation test only showed a very small drop of 1.3% in sensitivity at 100 °C after exposure of Bi-based sensors to the total neutron fluence of $2.5\times 10^{18}$ cm${}^{-2}$ in the LVR-15 reactor. The testing of Au-based sensors in neutron fluxes was carried out in the IBR-2 nuclear reactor to achieve a fluence of $10^{20}$ cm${}^{-2}$. The sensor sensitivity to magnetic field remained stable. The GR/Al${}_{2}$O${}_{3}$ Hall sensors have shown stability of their structure up to fluences of $4\times 10^{16}$ cm${}^{-2}$.

The great advantage of thin-film structures based on InSb is the possibility to precisely control their properties through appropriate doping. Determining the optimal initial value of *n* is very important to obtain adequate resistance to radiation under given irradiation conditions. However, it may be that the initial carrier densities optimal for the radiation resistance are not optimal for the thermal stability, still, they offer very good sensitivity to the magnetic field, incomparable to thin metallic layers. InSb can show excellent thermal stability up to 350 °C but at the cost of a decrease in the magnitude of $S_{I}$ and the resistance to radiation (suboptimal carrier density). The radiation resistance demonstrated by InSb is significant for the radiation dose in the range of $1.3\times 10^{18}$ cm${}^{-2}$ but is incomparably worse than in the case of the metallic layers of Bi and Au.

Experimental data on the radiation resistance of graphene [16,17,50] show that both transferred graphene and epitaxial graphene can exhibit significant radiation resistance. However, QFS graphene on SiC has incomparable temperature properties in comparison to those of the transferred one. This advantage manifests itself in extremely low thermal coefficients of both the sensitivity and the sheet resistance in the temperature range from liquid nitrogen to 500 °C. The transferred graphene is not a thermally stable system. $R_{S}$ changes close to 50% when the temperature changes by 20 °C [17]. Additionally, transferred graphene has a weaker adhesion to the dielectric layers than to the metal. Thermal compression of the upper contacts, which occurs after metal deposition and/or welding of external leads, can further weaken the graphene adhesion [17]. The technology of transferred graphene is not sufficient for the needs of devices intended to operate in extreme conditions, taking into account their temperature factors.

## 4. Conclusions

This work demonstrates the first experimental comparison of classic thin-film Hall-effect systems with a modern 2D system in the context of extreme-environment applications. The comparison is only possible if one defines the extreme operating conditions in the same manner for both of the investigated systems. Nine InSb-based thin films and an epitaxial GR/SiC system were exposed to high temperatures of up to 350 °C and neutron irradiation with fluence ≈$7\times 10^{16}$ cm${}^{-2}$ in the MARIA research nuclear reactor. The samples were exposed to neutron radiation at a high flux rate, up to $2.0\times 10^{12}\;{\mathrm{cm}}^{-2}s^{-1}$, which caused the thin films to warm to a temperature of about 200 °C.

Both investigated sensor platforms are perfectly suited to high-temperature applications, as they have high values of $S_{I}$ and high stability of $S_{I}$ in a wide range of temperatures. However, it is significant that the temperature limit that should not be exceeded for the thin-film InSb is the range of 400–500 °C, which is not a limitation for the GR/SiC system.

In terms of the effect of NR on inSb thin films, the results showed the creation of complex types of defects in thin films, of both donor and acceptor nature. It can be concluded that the formation rate of both types of defects is the same for thin-film electron densities of approximately (3.0–3.7) $\times 10^{17}\;{\mathrm{cm}}^{-3}$. This value can be considered optimal for the construction of a Hall device based on InSb for application in an environment containing mainly fast neutrons and for the neutron dose that does not exceed $7\times 10^{17}$ cm${}^{-2}$.

With the best initial carrier density value of the InSb structures (sample *I5*), the value of this parameter changed only by 22%, with a 39% change for GR/SiC after irradiation. This suggests a similar degree of signal stability for both systems; however, using precise doping for the InSb platform (optimizing the radiation stability of the system), it seems possible to go down to even lower signal loss values at a given radiation dose. Regardless of the doping level of InSb thin films, the charge carrier mobility for all samples decreased by an average of 83% (the lowest recorded decrease was 75%, and the highest was 90%).

Such drastic changes suggest significant structural damage. The decrease in the charge carrier mobility of the GR/SiC system was recorded by almost half, at the level of 43%. This is clearly due to the 2D active layer, in the case of which a very small probability value of a direct collision of a neutron with a graphene atom is provided. In this case, the main reason for the decrease in the mobility of carriers after the GR/SiC irradiation should be damage to other components of the system, rather than the defect of the active layer itself.

Post-NR temperature treatment showed that it is possible to remove some of the radiation defects. In the case of the GR/SiC system, changes in carrier density after annealing at temperatures up to 350 °C reached only a few percent. The accompanying improvement in carrier mobility did not exceed 10% of the initial value of this parameter. These effects are correlated with the surface diffusion of hydrogen atoms of the damaged intercalating layer of the GR/SiC system [16]. This effect brings a certain degree of self-healing ability to the GR/SiC system, but has no real effect on the reconstruction of the graphene active layer.

Thin films are much more susceptible to thermal regeneration. It has been found that the optimal upper limit of the temperature treatment of InSb structures is different depending on the type of substrate used. Epitaxial InSb layers on a monocrystalline GaAs substrate are more susceptible to higher annealing temperatures. Epitaxial InSb thin films epitaxially grown on the monocrystalline GaAs substrate regained the highest carrier mobility after the last applied heating cycle to 350 °C. Thin films deposited on the polycrystalline substrate recovered most of μ after the penultimate cycle to 300 °C. Therefore, the best films for this purpose could be thin films with a large number of crystal lattice defects, which will cause the defects introduced by NR to be less important.

However, thermal regeneration may not be as relevant as radiation hardness when operating under high levels of neutron radiation. Hence, thermally stable systems based on a 2D active layer seem to be a very promising solution to the problem of a magnetic field sensor resistant to neutron radiation. However, the GR/SiC system requires more extensive research to investigate its range of applicability.

## Figures and Tables

**Figure 1 sensors-22-05258-f001:**
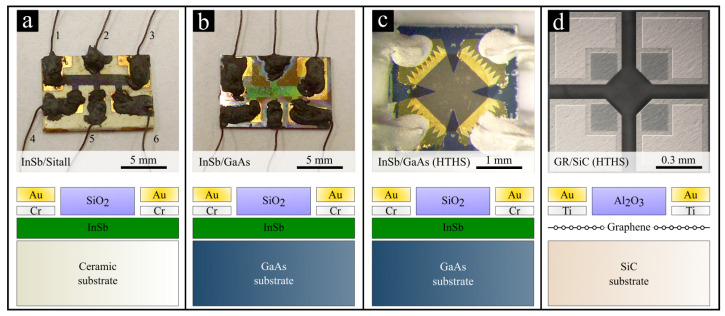
Schematics and optical images of the fabricated Hall effect structures (**a**) InSb/Sitall (**b**) InSb/GaAs (**c**) n-InSb (HTHS) (**d**) QFS GR/SiC (HTHS).

**Figure 2 sensors-22-05258-f002:**
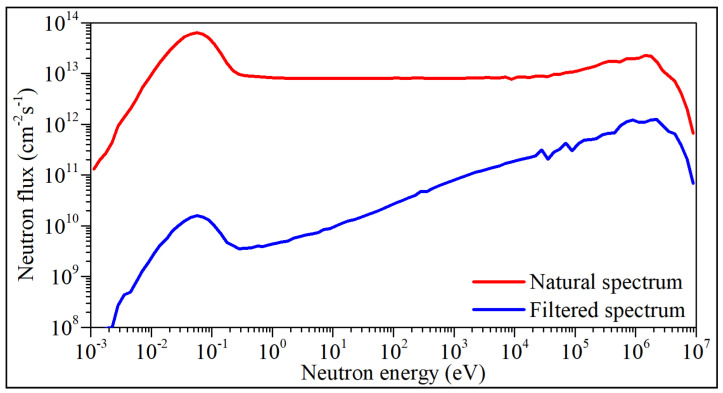
Natural and filtered energy spectra of the MARIA nuclear research reactor.

**Figure 3 sensors-22-05258-f003:**
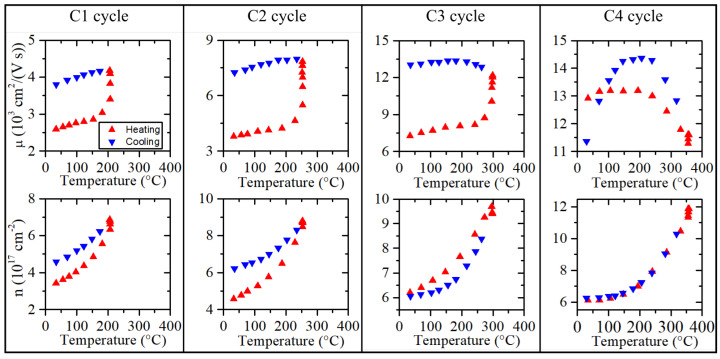
Complete temperature treatment process of the *I6* structure.

**Figure 4 sensors-22-05258-f004:**
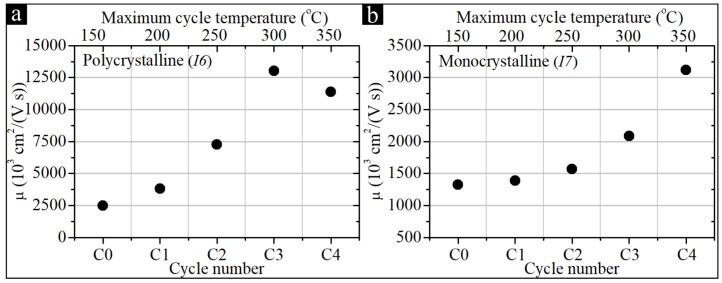
Summary of the final parameters at RT after each successive cycle of the temperature treatment procedure of structures *I6* (**a**) and *I7* (**b**).

**Figure 5 sensors-22-05258-f005:**
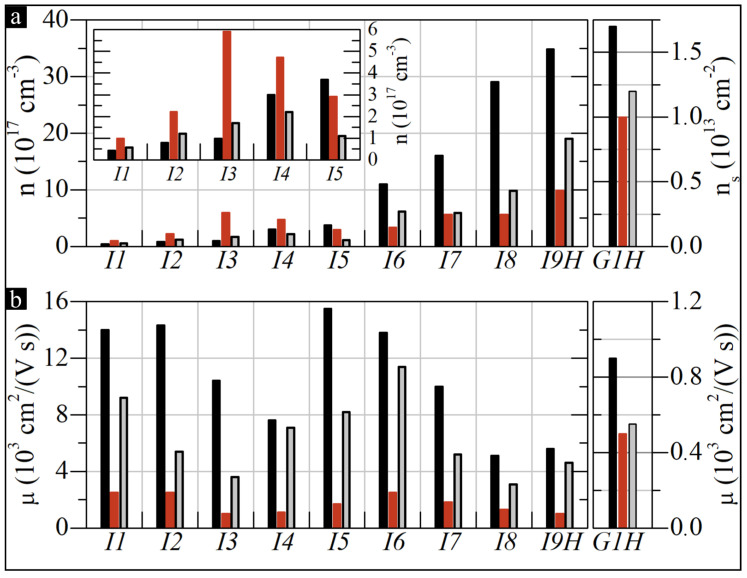
Charge carrier density (**a**) and mobility (**b**) in InSb and GR/SiC structures pre-NR, post-NR, and after the complete temperature treatment.

**Figure 6 sensors-22-05258-f006:**
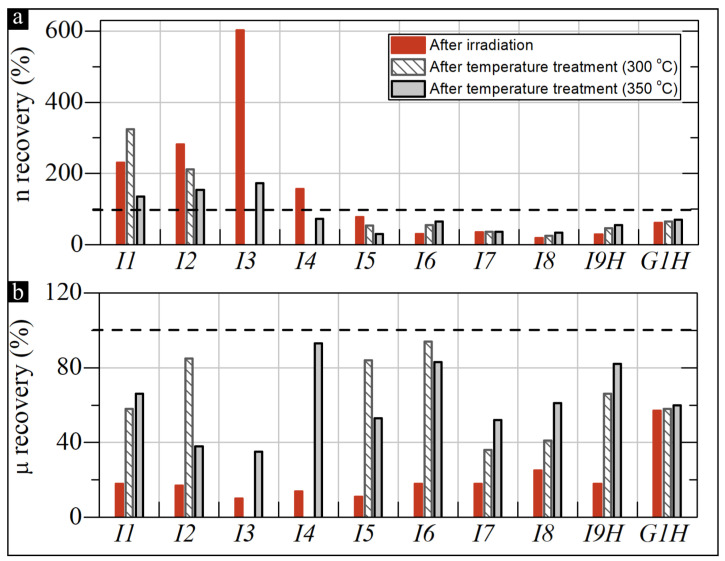
Recovery of charge carrier density (**a**) and mobility (**b**) in InSb and GR/SiC structures after the post-NR annealing cycles up to 300 °C and 350 °C.

**Figure 7 sensors-22-05258-f007:**
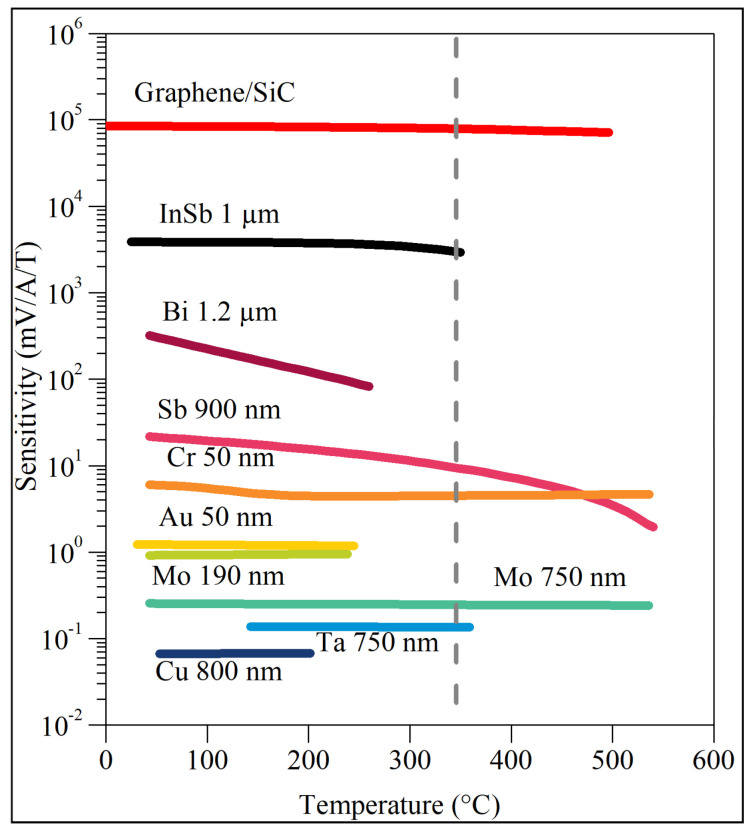
The sensitivity chart of GR/SiC and InSb-based structures in a wide temperature range. The sensitivity waveforms of other structures are based on the data from Refs. [5,42] and are shown for comparison.

**Table 1 sensors-22-05258-t001:** Summary of the electrical parameters of the verified structures after the thermal stabilization (*A*), after the irradiation (*B0*), after the temperature treatment up to $300{\;}^{{}^{\circ}}$C (*B1*), and after the temperature treatment up to $350{\;}^{{}^{\circ}}$C (*B2*).

InSb Sample No. <Dopant> Substrate	Film Thickness d, μm		Carrier Mobility μ, ${\mathrm{cm}}^{2}/\left(\mathrm{Vs}\right)$	$\left(\mu_{\mathrm{Bi}}-\mu_{A}\right)/\mu_{A},$ %	Carrier Density *n* $10^{17}{\mathrm{cm}}^{-3}$	$\left(n_{\mathrm{Bi}}-n_{A}\right)/n_{A},$ %
I1 <-> GaAs	4.5	A	14,000	-	0.42	-
		B0	2500	−82	0.97	131
		B1	8100	−42	1.36	224
		B2	9200	−34	0.57	36
I2 <-> Sitall	2.0	A	14,300	-	0.78	-
		B0	2500	−83	2.20	182
		B1	12,200	−15	1.65	112
		B2	5400	−62	1.20	54
I3 <-> Sitall	2.0	A	10,400	-	0.98	-
		B0	1000	−90	5.90	502
		B1	no measurement	-	no measurement	-
		B2	3600	−65	1.70	73
I4 <Te> Sitall	1.7	A	7600	-	3.00	-
		B0	1100	−86	4.70	57
		B1	no measurement	-	no measurement	-
		B2	7100	−7	2.20	−27
I5 <Te> Sitall	2.0	A	15,500	-	3.70	-
		B0	1700	−89	2.90	−22
		B1	13,000	−16	2.05	−45
		B2	8200	−47	1.10	−70
I6 <Se> Sitall	3.1	A	13,800	-	11.00	-
		B0	2500	−82	3.30	−70
		B1	13,000	−6	6.06	−45
		B2	11,400	−17	6.20	−44
I7 <Sn> GaAs	1.0	A	10,000	-	16.00	-
		B0	1800	−82	5.60	−65
		B1	3600	−64	5.83	−64
		B2	5200	−48	5.90	−63
I8 <Sn> GaAs	1.1	A	5100	-	29.0	-
		B0	1300	−75	5.60	−81
		B1	2100	−59	7.33	−75
		B2	3100	−39	9.80	−66
I9H <Sn> GaAs	1.0	A	5600	-	34.80	-
		B0	1000	−82	9.80	−71
		B1	3700	−34	16.20	−53
		B2	4600	−18	19.00	−45
GR/SiC sample no. substrate			carrier mobility μ, cm${}^{2}$/(Vs)	$\left(\mu_{Bi}-\mu_{A}\right)/\mu_{A},$ %	sheet carrier density *n* 10${}^{13}$ cm${}^{-2}$	$\left(n_{Bi}-n_{A}\right)/n_{A},$ %
G1H <monolayer> 4HSiC		A	900	-	1.70	-
		B0	500	−43	1.00	−39
		B1	520	−42	1.12	−34
		B2	550	−40	1.20	−30

**Table 2 sensors-22-05258-t002:** Summary of the most important results for the NR resistance of materials for the construction of Hall sensors.

Material	Reactor Type	Neutron Fluence ${\mathrm{cm}}^{-2}$	Optimal *n* ${\mathrm{cm}}^{-3}$	$S_{I}$ Change, %	$T_{\mathrm{irr}}$ °C
InSb [43,44]	IBR-2	$10^{14}$	(1–3) $\times 10^{17}$	0.12	25
InSb [45]	IBR-2	(1.1) $\times 10^{15}$	6.7 $\times 10^{17}$	<1	17
InSb [46]	IBR-2	$10^{14}$–$10^{16}$	$2.5\times 10^{18}$	0.03–1	<70
InSb [45,47]	IBR-2	(0.7–3.1) $\times 10^{16}$	6.4 $\times 10^{17}$	<1	17
InSb [48]	IBR-2	$3.1\times 10^{16}$	(6–7) $\times 10^{17}$	0.01	
InSb [49]	WWR-2	$7.4\times 10^{17}$	(4–6) $\times 10^{17}$	<40	110
InSb [14]	MARIA *	(0.9–1.2) $\times 10^{18}$	n/c	100	<200
InSb [15]	LVR-15	$1.1\times 10^{17}$	n/c	<3	n/c
InSb [45]	LVR-15	$\left(2.5\right)\times 10^{17}$	n/c	6–23	90
InSb [45]	LVR-15	(0.3–1.3) $\times 10^{18}$	n/c	7–30	90
Bi [9]	LVR-15	$2.5\times 10^{18}$	n/c	1.3	100
Sb [11,13]	-	n/c	n/c	n/c	n/c
Cr [5]	-	n/c	n/c	n/c	n/c
Au [50]	IBR-2	$10^{16}$–$10^{20}$	n/c	<3	130
Mo [42]	-	n/c	n/c	n/c	n/c
Ta [42]	-	n/c	n/c	n/c	n/c
Cu [42]	-	n/c	n/c	n/c	n/c
Graphene ** [17]	IBR-2	(0.1–1.5) $\times 10^{16}$	n/c	3	55

* The natural spectrum of MARIA reactor; ** graphene transferred to sapphire; n/c—not considered.

## Data Availability

The raw/processed data required to reproduce the above findings cannot be shared at this time as the data also form part of an ongoing study.

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
