# Peer review of "The Comparison of InSb-Based Thin Films and Graphene on SiC for Magnetic Diagnostics under Extreme Conditions"

_sensors, 2022, doi:10.3390/s22145258_

Round 1

Reviewer 1 Report

The presented manuscript is well written where experimental results are nicely presented but the readability of the paper should be improved. Overall, this manuscript can be accepted.

Reviewer 2 Report

1. Abstract section doesn't provide information about the problem selected. The abstract section must be in standard form.

2. Organization of the manuscript should be checked such as Literature Review and Problem Formulation should be a separate section. 

3. Author should provide a comparative analysis between the proposed work and with the existing method

4. Proposed technique should be explained with novelty

5. Contribution of the author should be provided in the manuscript.

6. Results section should be explained comprehensively with positive points of the proposed work.

7. Equation parameters are not described properly.

8. Conclusion section should include the numerical results of the proposed work

9. What high temperature range was chosen for the sensor platform investigation?

10. Why is the optimal upper limit of InSb structure temperature treatment different depending on the type of substrate used?

11. What are those extreme conditions in which the comparative test study has been conducted. It must be clearly mentioned in the abstract section. 

12. In Figure 6, on what basis is the range of the base axis selected? What about the other ranges?

Reviewer 3 Report

The application of H-intercalated Graphene on SiC as a Hall sensor in radiation-environment is a very brilliant idea. The comparatory experiment results indeed show the potential advantage of such system in related applications.

Just a minor correction in Table I. The carrier mobility of sample I1 seems to be mis-typed (14000, instead of 1400).
